# Treatment of Painful Diabetic Neuropathy—A Narrative Review of Pharmacological and Interventional Approaches

**DOI:** 10.3390/biomedicines9050573

**Published:** 2021-05-19

**Authors:** Mayank Gupta, Nebojsa Nick Knezevic, Alaa Abd-Elsayed, Mahoua Ray, Kiran Patel, Bhavika Chowdhury

**Affiliations:** 1Kansas Pain Management & Neuroscience Research Center, Overland Park, KS 66201, USA; mahoua.r@kansaspainmanagement.com; 2Department of Anesthesiology, Advocate Illinois Masonic Medical Center, Chicago, IL 60657, USA; nick.knezevic@gmail.com; 3Department of Anesthesiology, University of Wisconsin School of Medicine and Public Health, Madison, WI 53715, USA; abdelsayed@wisc.edu; 4Department of Pain Management, Spine and Pain Institute of New York, New York, NY 10065, USA; kiranpatelmd@yahoo.com; 5Department of Endocrinology, Saint Luke’s South Hospital, Overland Park, KS 66213, USA; Bhavikachowdhury@hotmail.com

**Keywords:** painful diabetic neuropathy, diabetes, neuropathic pain, peripheral diabetic neuropathy, neuromodulation, 10 kHz SCS, spinal cord stimulation

## Abstract

Painful diabetic neuropathy (PDN) is a common complication of diabetes mellitus that is associated with a significant decline in quality of life. Like other painful neuropathic conditions, PDN is difficult to manage clinically, and a variety of pharmacological and non-pharmacological options are available for this condition. Recommended pharmacotherapies include anticonvulsive agents, antidepressant drugs, and topical capsaicin; and tapentadol, which combines opioid agonism and norepinephrine reuptake inhibition, has also recently been approved for use. Additionally, several neuromodulation therapies have been successfully used for pain relief in PDN, including intrathecal therapy, transcutaneous electrical nerve stimulation (TENS), and spinal cord stimulation (SCS). Recently, 10 kHz SCS has been shown to provide clinically meaningful pain relief for patients refractory to conventional medical management, with a subset of patients demonstrating improvement in neurological function. This literature review is intended to discuss the dosage and prospective data associated with pain management therapies for PDN.

## 1. Introduction

The number of patients living with diabetes mellitus (DM) is growing in the United States, and the estimated prevalence of this condition rose from 9.5% in 1999–2000 to 12.0% in 2013–2016 [1]. Neuropathy, which can produce both painful and non-painful symptoms, is the most common complication of DM [2,3], and the management of neuropathic symptoms has been estimated to comprise 27% of the total annual cost of diabetes care or 9% of all healthcare costs for people with DM [4]. Painful diabetic neuropathy (PDN) has been estimated to affect 20% to 24% of all patients with DM [5], and PDN has been reported in 19% of those with insulin-dependent DM and 49% of those with non-insulin-dependent DM [6]. Pain has also been reported in twice as many patients with DM and neuropathic symptoms (60%) than those with DM but no neuropathy (30%) [7].

Like other chronic pain conditions, PDN has been associated with substantial declines in quality of life measures including sleep, recreational activities, normal mobility, general activity, social activities, and mood [8]. Like neuropathic pain from all etiologies, PDN is often refractory to treatment and challenging to treat [9], and a variety of strategies are currently employed to manage this condition (Table 1). Because no single treatment modality is beneficial or appropriate for every patient with PDN, it is undoubtedly desirable to have multiple options for managing this condition; however, some of the evidence supporting different options varies substantially. The pivotal studies supporting the use of current treatments for PDN, including pharmacotherapies and neuromodulation, are presented here with a focus on results regarding their effectiveness and safety. These results reveal important differences in what is and is not known about these treatments and their adverse events (AEs), which can help guide clinicians in choosing the best option.

## 2. Clinical Presentation and General Management

The most common clinical presentation of diabetic neuropathy is distal symmetric polyneuropathy (DSP) [10,11], which affects the limbs symmetrically in a characteristic “glove and stocking” pattern. Common symptoms of DSP include numbness, tingling, and weakness, in addition to pain, and many patients experience sensations similar to bunched-up socks or ill-fitting shoes [10]. Subjects with PDN have described the pain as ‘burning’, ‘electric’, ‘sharp’, and ‘dull/ache’, and the intensity of pain has been reported to worsen in about half of those with PDN at night, when tired, or when stressed [8].

PDN is a diagnosis of exclusion; therefore, a combination of a thorough medical history, clinical testing, and neurological examination is required to eliminate possible secondary causes of pain [12]. It is important to closely monitor patients with DM for neuropathic symptoms, since over 12% of patients in one study did not report painful symptoms to their physicians [13]. Glycemic control can help to help prevent or slow the progression of PDN, but there is currently no available treatment to reverse existing nerve damage [14]. This means symptoms, including pain, will need to be managed chronically and a therapy that is both effective and safe in the long-term is required. In the following sections, different types of treatments, including pharmacological as well as non-pharmacological therapies that are currently available or under evaluation, are summarized.

## 3. Criteria for Selection of Articles

Pivotal randomized studies that met the following criteria were identified through literature search and discussed in the review. The qualification criteria included:(a)Studies that evaluated the safety and efficacy of the drug or device.(b)Studies that supported the new drug application (NDA) or the post market follow-up requirements.

Where indicated, additional information was obtained from the FDA-approved package insert and discussed in the review.

## 4. Pharmacotherapy

The first treatment for pain, including chronic pain, is often pharmacotherapy, but neuropathic pain is not the same as musculoskeletal pain, and commonly used analgesics such as opioids are not appropriate or effective for managing chronic neuropathic pain such as PDN [15,16]. The pharmacotherapies approved and used to manage PDN are mostly not traditional analgesics or opioids that can be taken “as needed” but rather agents such as anticonvulsants or antidepressants that must be taken regularly for a period of time to achieve full effect [17]. There are now more pharmacotherapies available for treating PDN than in the recent past, which are summarized in Table 2, and clinicians should consider patient-specific factors such as age, quality-of-life goals, functional status, and comorbidities when determining appropriate management [3,18,19,20].

### 4.1. Anticonvulsants

Pregabalin and gabapentin are both gabapentanoids that act as anticonvulsive drugs by inhibiting α_2_-δ calcium channels in the dorsal horn, thereby inhibiting neurotransmitter release [23,24,25,26,27]. Pregabalin has been approved by the U.S. Food and Drug Administration (FDA) for use in treating PDN, and is recommended by the American Diabetes Association (ADA) as a first-line treatment [18,23], and while gabapentin is not approved for this indication, it is also recommended by the American Academy of Neurology (AAN) and ADA for this use [18,19].

The efficacy of pregabalin was shown in a double-blind, parallel-group, randomized controlled trial (RCT) in subjects with PDN [28]. A total of 146 subjects were randomized to treatment with 300 mg/day pregabalin or placebo for 8 weeks, and treatment with pregabalin was associated with a 38% decrease in pain scores from baseline, which was significantly more than the 13% reported in placebo-treated subjects (Table 3). The most frequently reported AEs were reported more often in the treatment arm than placebo and included dizziness, somnolence, infection, and peripheral edema [28]. More pregabalin-treated subjects discontinued treatment due to AEs (11%) than those treated with placebo (3%), but more subjects in the placebo arm discontinued due to lack of efficacy (4%) than in the treatment arm (1%). The FDA approved dosage for treatment of PDN is 50 mg TID (150 mg/day) at initiation and can be titrated up to 100 mg TID (300 mg/day) to achieve adequate effect [23]. AEs reported with pregabalin treatment across many clinical studies include somnolence, dizziness, blurred vision, difficulty with concentration or attention, dry mouth, edema, and weight gain [25].

Gabapentin is another anticonvulsant frequently used to treat PDN, although it is characterized as a second-line alternative to pregabalin due to the lower quality of clinical data available for gabapentin, its less predictable pharmacokinetics, longer titration periods, less flexible dosing, and requirement for dosing adjustments in patients with renal impairments [20]. Dosing for chronic pain starts at 300 mg/day and is titrated up until suitable pain relief is achieved with effective doses ranging from 1800 mg to 3600 mg per day [27]. Gabapentin has been tested for the treatment of PDN in a multi-center, double-blind RCT that enrolled 165 subjects [29]. Subjects were initiated at a dose of 900 mg/day gabapentin and were titrated to a maximum of 3600 mg/day. Mean pain relief in the treatment arm was 39%, significantly greater than the 22% decrease reported in the placebo arm after 8 weeks of treatment, and the most common AEs were sedation and dizziness [29]. AEs reported across a range of clinical studies using gabapentin to treat neuropathic pain include dizziness, somnolence, peripheral edema, and gait disturbance [35].

### 4.2. Antidepressants

Antidepressants including serotonin and norepinephrine reuptake inhibitors (SNRIs) like duloxetine and venlafaxine treat chronic neuropathic pain by increasing the activity of noradrenergic and serotonergic neurons in the descending pathways of the dorsal horn. These descending neurons inhibit the activity of dorsal horn neurons, suppressing excessive input, which is perceived as pain, from reaching the brain [36,37]. Tricyclic antidepressants (TCAs), likewise, block monoamine reuptake, including serotonin and norepinephrine, and are also used to treat chronic pain, especially neuropathic pain [37,38].

Duloxetine became the first agent approved by the FDA for treating PDN in 2004 [39], and is recommended as a first-line treatment for neuropathic pain by the AAN and the ADA [18,19]. The first of two pivotal trials of this agent was a multicenter, parallel, double-blind RCT that tested duloxetine at 60 mg or 120 mg/day for 12 weeks against placebo treatment in 348 subjects and reported mean pain reductions of 64% to 68% in the treatment groups, which were both significantly higher than the 43% mean pain reduction reported in placebo-treated controls. Treatment-emergent AEs (TEAEs) including nausea, somnolence, hyperhidrosis, and anorexia were more common in both duloxetine-treated groups than among placebo-treated controls, and the most frequently cited reasons for treatment discontinuation were vomiting and nausea [30]. The second pivotal trial was, likewise, a multicenter, double-blind RCT in 457 subjects with PDN that tested duloxetine doses of 20 mg, 60 mg, and 120 mg per day [40]. Mean pain relief after 12 weeks of treatment was significantly greater than that observed in placebo-treated subjects (33%) in the 60 mg/day (48%) and 120 mg/day (54%) treatment groups. TEAEs that were significantly more common with 120 mg/day duloxetine treatment than placebo include constipation, dry mouth, hyperhidrosis, decreased appetite, anorexia, weakness, nausea, and severe somnolence. The recommended dosage of duloxetine for PDN is 60 mg/day, and lower initial doses may be used in cases with tolerability concerns or renal impairment, which is a common complication of diabetes. The duloxetine label includes a black box warning for the potential emergence or worsening of suicidal thinking or behavior in children and young adults, while frequently reported AEs include nausea, dry mouth, somnolence, constipation, decreased appetite, and hyperhidrosis [40].

In addition to duloxetine, the SNRI venlafaxine and TCAs have shown evidence of efficacy for PDN and may be considered for PDN according to recommendations by the ADA, although they have not received FDA approval for this use [18]. Venlafaxine is mechanistically similar to duloxetine, but there are fewer published data for this drug in PDN [17]. An extended release (ER) formulation of venlafaxine was tested in a 6-week, double-blind, RCT in 244 subjects with PDN, and the investigators found that doses from 150 mg to 225 mg/day resulted in 50% lower pain scores than the baseline, which was significantly greater than the 27% pain reduction in the placebo group [41]. The reported AEs in this trial were nausea, somnolence, and electrocardiogram abnormalities, and there was no significant difference in the rate of serious AEs between subjects receiving placebo (10%) or 150–225 mg/day venlafaxine ER (12%).

Amitriptyline is the most commonly used TCA for treating PDN, but the high risk of AEs requires careful monitoring and this drug is best suited as a last resort [17]. A meta-analysis of trials testing amitriptyline in subjects with PDN included four studies with doses ranging from 10 mg to 90 mg/day for 12 to 14 weeks [34]. The authors concluded amitriptyline was 1.95-fold more effective than placebo at producing at least 50% pain relief, which was not significant, but the odds ratio (OR) for subject withdrawals due to AEs for amitriptyline was 10.24 relative to placebo and 7.03 relative to gabapentin [34]. Common AEs encountered with TCAs include gastrointestinal issues, orthostatic hypotension, dry mouth, urinary retention, and QTc prolongation, and this safety profile reflects concurrent actions at histaminergic, adrenergic, and cholinergic receptors [17].

### 4.3. Opioids

Opioids have been commonly used in the past for treating chronic non-cancer related pain [16]; however, there is little evidence that opioids effectively reduce chronic pain, including neuropathic pain [42,43]. In addition, the serious risks presented by long-term opioid use include respiratory depression and addiction [44], which have led to an emphasis on avoiding their long-term use, whenever possible [15].

Currently, tapentadol is the only opioid specifically approved by the FDA for use in treating PDN. Tapentadol is a strong analgesic that combines the mechanisms of a µ-opioid receptor agonist, like a typical opioid, and a norepinephrine reuptake inhibitor [45]. The opioid effects inhibit ascending pain signals in the spine, while increased synaptic levels of norepinephrine potentiate descending inhibitory signaling. Tapentadol was approved for use with PDN in 2012 [46], but the dangers of chronic opioid treatment make the use of this drug for PDN controversial.

The efficacy of tapentadol in treating neuropathic pain was documented in two nearly identical phase 3 trials, which randomized 713 subjects in total [47,48]. Both studies started with a three-week open-label stage, during which subjects with PDN were titrated to an optimal dose of tapentadol (200 mg/day–500 mg/day) for pain management, and all subjects who responded to tapentadol were randomized to continue on tapentadol treatment or placebo in a blinded manner. Schwartz et al. reported a 37% increase in pain scores in subjects switched to placebo and no change in those continuing on tapentadol, a significant difference [47], while the RCT by Vinik et al. showed subjects who continued on tapentadol treatment in the double-blind part of the study also had significantly less pain (26%) than those who were withdrawn to placebo [48]. TEAEs led to discontinuation of tapentadol in 17% and 20% of subjects during the open-label portions of these studies, and the primary causes of discontinuations were nausea, vomiting, and dizziness [47,48]. Since both study populations were enriched for patients who responded to tapentadol, the applicability of these results to the general population is debated. This fact, combined with safety concerns about chronic opioid intake, prompted the ADA not to recommend this medication as a first- or second-line treatment of PDN [18]. Due to the addictive nature of opioids, it is advised to avoid prescribing tapentadol for PDN and consider alternate options for the management of pain [32].

### 4.4. Topical Capsaicin

In 2020, the FDA approved a capsaicin 8% topical patch for treating PDN [49]. Topical capsaicin has been used for treating pain with a variety of etiologies and works via agonism of the transient receptor potential vanilloid 1 receptor (TRPV1). Topical exposure is thought to reduce the TRPV1-expressing nociceptive nerve endings in the affected area, providing a period of pain relief lasting several months [49]. The efficacy of topical capsaicin in reducing pain due to PDN was demonstrated in a 12-week, double-blind RCT study in 369 subjects, showing that mean pain was reduced by 28% in the treatment group and by 21% in the placebo group, which was statistically significant [50]. Only three subjects in this study, all in the 8% capsaicin treatment group, had severe drug-related TEAEs, including two with severe burning sensations and one with severe application site pain, but no subjects discontinued due to drug-related TEAEs.

## 5. Non-Pharmacological Treatments: Neuromodulation

The International Neuromodulation Society defines neuromodulation as medical technologies that reversibly enhance or suppress nervous system activity with the goal of treating disease and includes both implantable and non-implantable devices that deliver electrical, chemical, or other agents [31]. Types of neuromodulation tested in subjects include transcutaneous electrical nerve stimulation (TENS), intrathecal pain therapy, and spinal cord stimulation (SCS). Although such methods have been used with well-documented success in conditions such as musculoskeletal pain or failed back surgery syndrome, few well-controlled studies have examined their use in PDN. However, there is increased interest in new, non-opioid methods for treating neuropathic pain, and the use of neuromodulation is expected to expand in the coming years [44]. Several neuromodulation methods with positive published results are reviewed here and include varying levels of invasiveness and efficacy, and this information is summarized in Table 4.

### 5.1. Transcutaneous Electrical Nerve Stimulation

TENS is a non-invasive, inexpensive, and easy-to-use form of neuromodulation to treat both acute and chronic pain with few contraindications or AEs, and no known drug interactions [66]. Patients treated with TENS have electrical stimulation applied to the skin via adhesive electrodes using a variety of waveforms that are broadly classified as high frequency (>50 Hz), low frequency (<10 Hz) or burst. The mechanism by which TENS produces its analgesic effects is currently unknown, but multiple complimentary hypotheses have been proposed including improved microcirculation, higher levels of beta endorphin and met-enkephalin, increased expression of proteins including calcitonin gene regulating protein and nerve growth factor, and reduced inflammation [67].

Despite its long history of clinical use, there is no consensus on the efficacy of TENS for treating pain [68], and trials of this treatment in neuropathic pain have tended to be small, short in duration, and with large placebo effects, making the utility of TENS for managing PDN uncertain [67,69]. A pair of small trials in the 1990s showed improvements in both pain and other neuropathic symptoms in patients with PDN when receiving TENS compared to sham controls [70,71]. In the first of these, TENS produced pain relief by 54% in 18 subjects and improvement of neuropathic symptoms in 15, significantly more than the 18% decline in pain scores and neurologic improvement in 5 of 13 sham-treated controls [71]. The second prospective, randomized study tested TENS in 14 subjects with PDN who had not responded to four weeks of treatment with amitriptyline. In this trial, TENS plus amitriptyline produced 66% pain relief, which was significantly more than the 55% pain relief in subjects receiving amitriptyline plus sham stimulation [70]. No AEs were reported in either trial among patients receiving TENS stimulation. An RCT that included 25 subjects with PDN compared three days of treatment with TENS (≤35 Hz) to treatment with high-frequency external muscle stimulation (>4 kHz), and the authors reported a significant reduction in “total symptom score” including pain with both treatments [51]. However, fewer subjects with painful PDN responded to TENS treatment (25%) than high-frequency external muscle stimulation (69%), with a response defined as alleviation of at least 1 symptom by 3 or more points on an 11-point scale [51].

Low-frequency pulsed electromagnetic fields (PEMF) was tested in 225 subjects with PDN, but the reported mean pain relief of 28% was not significantly different from placebo after a three-month RCT [52]. The only AE reported was allodynia leading to two subjects each from the PEMF and sham treatment groups to drop out of the study. Frequency-modulated electromagnetic neural stimulation (FREMS) was also tested in a long-term RCT in 110 patients with symptomatic diabetic neuropathy and produced a statistically significant effect, decreasing pain scores by about 50%, but this effect was transient, and undetectable three months after the last treatment [53].

### 5.2. Intrathecal Pain Therapy

Intrathecal pain therapy is a targeted drug delivery strategy to bypass first pass metabolism and the blood-brain barrier by delivering analgesic medication directly into the intrathecal cerebrospinal fluid via a pump and catheter to treat refractory chronic pain when conventional medical treatments are ineffective [72]. Intrathecal therapy using either ziconotide or morphine is recommended and FDA-approved for chronic neuropathic pain such as that associated with PDN [73,74]. Ziconotide is recommended more strongly by the Polyanalgesic Consensus Conference (PACC) because it is supported by evidence from well-designed trials, unlike the use of morphine, and it is not associated with some of the serious AEs observed with opioids, especially respiratory depression [75].

Mechanistically, intrathecal morphine acts as a µ-opioid agonist to reduce pain, as in other delivery methods, but targeted delivery to the spine can increase efficacy, improve alertness, and reduce AEs compared to systemic opioid therapy [76]. Intrathecal ziconotide, in contrast, is a nonopioid analgesic that binds selectively and reversibly to N-type voltage-sensitive calcium channels, thereby blocking the release of pro-nociceptive neurotransmitters in the spinal dorsal horn [76]. Tolerance and withdrawal do not develop in response to intrathecal ziconotide, which is a significant advantage over intrathecal morphine, and serious AEs are rare, even in cases of overdose. However, this agent requires careful dose titration, is contraindicated for patients with psychosis, and carries a black box warning for potential severe psychiatric symptoms and neurological impairment [73].

The efficacy of ziconotide has been demonstrated for chronic nonmalignant pain in a double-blind RCT involving 255 total subjects randomized 2:1 to treatment with ziconotide or placebo [54]. Over 75% of the subjects in the treated arm had chronic neuropathic pain, and the results showed significant reductions in pain scores among subjects treated with ziconotide relative to placebo during the six-day study period. Among patients receiving ziconotide, 95% experienced at least 1 AE, and AEs likely to be related to ziconotide included nausea, hypotension, dizziness, somnolence, urinary retention, asthenia, amblyopia, nystagmus, abnormal gait, and confusion [54]. Far fewer data from prospective trials exist for the efficacy and safety of intrathecal morphine; however, a small RCT was reported in subjects with non-cancer pain who had received intrathecal morphine for at least 1 year [77]. The investigators progressively reduced the morphine dose for subjects in the intervention arm, while maintaining the doses for those in the control group, and found increased pain scores and study discontinuations among those receiving reduced doses, demonstrating efficacy although the sample size was small. Serious AEs commonly reported in association with intrathecal morphine include respiratory depression that can lead to death, the formation of inflammatory masses (granulomas) around the catheter tip, and myoclonus [76].

### 5.3. Conventional SCS

Conventional, tonic SCS was first used to treat human pain in 1967 [78], and it has been established as a standard treatment for chronic, refractory pain since the 1980s [79]. Conventional SCS is administered with a variety of waveforms via electrode leads implanted in the epidural space. Typically, a frequency of 40 Hz with a pulse width of 400 µs is delivered at intensities high enough to produce paresthesia, which is necessary to produce analgesic effects and must overlap the painful area [80]. The current understanding of the mechanism by which conventional SCS produces analgesia is based, in part, on the Gate Control Theory by Melzak and Wall [81]. Stimulation through the epidural electrodes activates large diameter spinal Aβ fibers in the dorsal column of the spine, which is thought to produce both pain relief and paresthesia, and the intensity of stimulation is correlated with the inhibition of wide-dynamic range neurons in the dorsal horn [82]. In addition, functional MRI imaging of patients undergoing tonic SCS at conventional frequencies has shown the activation of supraspinal areas that modulate pain transmission in the dorsal horn via descending serotonergic and noradrenergic projections [83].

The first clinical study of conventional SCS in patients with chronic, refractory PDN included 10 subjects who had a trial stimulation and 8 who proceeded to a permanently implanted system [55]. The investigators reported significant relief of both background and peak neuropathic pain through 14 months of stimulation, and follow-up visits at 3.3 and 7.5 years found continued pain relief in these subjects [84]. The investigators reported several stimulation-related AEs including loss of analgesia in 1 subject, superficial wound infections, hematoma, electrode migration and displacement, and electrode failure secondary to trauma. The positive efficacy results were supported by two more small open-label prospective studies of SCS for treating PDN that reported positive responses in 9 of 11 subjects after 6 months of treatment [56], and 10 of 15 subjects after 12 months of stimulation [57].

Based on this work, two RCTs were conducted to obtain better evidence regarding the safety and efficacy of SCS for treating PDN refractory to conventional medical treatment. In the first of these, 60 subjects were randomized 2:1 to treatment with best conventional medical practice with or without SCS therapy and followed for six months [61]. The authors reported a significant reduction of 55% in pain scores in the group treated with SCS and no change in pain intensity among controls, and 60% of subjects treated with SCS (vs. 5% of controls) reported more than 50% pain relief, a common threshold used to define SCS responders. AEs included infections, pain at the implant site, and electrode lead migration. In the second RCT, 36 subjects were randomized to receive the best medical treatment with or without SCS, and the investigators reported pain relief of 44% during the day and 38% at night in those receiving SCS for 6 months, while controls reported 0% relief during the day and 10% at night [58]. The proportion of subjects reporting at least 50% pain relief after SCS treatment was 41% during the day and 36% at night. Serious AEs reported in this study were one death due to subdural hematoma and one infection resulting in explant and autonomic neuropathy. After six months, 93% of subjects in the control arm crossed over to the SCS treatment arm and 15 subjects in total were evaluable after 24 months of treatment [59]. The mean pain relief was 45% during the day and 48% at night, and the proportion of subjects experiencing at least 50% pain relief was 47% during the day and 35% at night. After 24 months of stimulation, 13% of subjects had undergone surgery to replace the implanted pulse generator (IPG), and 27% underwent lead revisions. Most recently, long-term results for subjects from this RCT plus those from the pilot study by Pluijms et al. [57] have been published, showing that among 48 subjects treated for a median of 5 years, mean pain relief decreased to 36% during the day and 31% at night, while the proportion of subjects experiencing at least 50% pain relief decreased to 36% and 32% during the day and night, respectively [60].

Despite the benefits of conventional tonic SCS for some patients with PDN, many patients do not respond to this treatment, and physiological adaptation, or habituation, frequently results in a loss of therapeutic effect [85]. Moreover, paresthesia itself is a limiting factor for a substantial number of patients, particularly paresthesia affecting areas outside of the painful region and carrying intensity variations resulting from postural changes [86]. Alternative waveforms have been developed with the aim of addressing some of these limitations and have been studied in patients with PDN, including burst SCS and high-frequency SCS [62,63,64,87,88].

### 5.4. Burst SCS

Unlike conventional and high-frequency SCS, which deliver stimulation at a constant, or tonic, frequency, burst SCS is characterized by clusters of high frequency pulses separated by longer inter-pulse intervals and is intended to emulate naturally occurring neuronal firing patterns [89]. Like low-frequency tonic SCS, burst SCS produces analgesia via GABAergic mechanisms, and intrathecal administration of GABA_A_ and GABA_B_ antagonists abolish the analgesic effects of both types of SCS [90]. Brain imaging in humans has also shown supraspinal effects of burst SCS that activates areas involved with emotion and motivation to a greater extent than tonic, low-frequency SCS [83].

Limited evidence for the efficacy of burst SCS in PDN was shown by a prospective trial of subjects with at least six months of previous experience with conventional SCS that included 12 subjects with PDN [87]. Subjects with PDN reported additional pain reduction averaging 44% after two weeks of burst stimulation, which was significant, and eight (67%) preferred burst SCS to conventional stimulation. AEs reported among all 48 subjects in this brief study were headaches, dizziness, and the sensation of “heavy legs”, and several reported feeling paresthesia in the supine position [87].

### 5.5. High Frequency (or 10 kHz) SCS

In contrast to conventional and burst SCS, evidence from a multicenter RCT has shown that high-frequency SCS delivered at 10 kHz (10 kHz SCS) produces deep and durable paresthesia-free pain relief for chronic neuropathic pain [44,62,63,64,88,91,92,93,94,95,96,97]. The specific benefits and unique physiological responses associated with 10 kHz SCS may be attributed to its unique mechanism of action [98,99,100,101,102,103,104,105], although the precise mechanism of action is not yet understood [106].

Research in rodents using in vivo and ex vivo electrophysiological methods has shown that sub-sensory threshold stimulation at 10 kHz selectively activated inhibitory interneurons in the spinal dorsal horn, unlike such stimulation delivered at 1 kHz or 5 kHz, suggesting that low-intensity 10 kHz SCS may produce paresthesia-free pain relief by activating inhibitory interneurons in the spine without activating dorsal column fibers [98]. A study using a spared nerve injury-induced (SNI) neuropathic pain model in rats showed that 10 kHz SCS significantly reduced hyperalgesia compared to sham stimulation and was also associated with reduced levels of inflammatory mitogen-activated protein kinases (MAPKs) in the dorsal root ganglia [99]. Alterations in glutamatergic signaling in the dorsal horn has been shown to be involved in the development of neuropathic pain in rodents [107], and results from a second study—a rat SNI pain model—demonstrated that 10 kHz SCS relieved pain and partially restored altered spinal glutamate uptake activity, spinal glutamate levels, and miniature excitatory postsynaptic currents [108].

Work in humans has produced other possible mechanisms of action. Investigators who recorded 10-channel electroencephalograms in nine patients during SCS surgery reported a shift in peak frequencies from theta at baseline or with tonic stimulation at 60 Hz to alpha rhythms with high frequency stimulation at 1 or 10 kHz [103]. In addition, the authors reported a positive correlation between disability scores and the high-frequency stimulation-induced alpha/theta peak power ratio in patients’ frontal and somatosensory brain regions. Investigators who examined human subjects using voxel-based morphometry reported that in subjects with pain due to failed back surgery syndrome who were treated with 10 kHz SCS, pain relief was correlated with bilateral decreases in hippocampal volume, demonstrating an effect of this therapy on structural brain architecture over time [105].

A prospective trial, SENZA-PPN, was the first to test 10 kHz SCS in 26 subjects with peripheral polyneuropathy (PPN) refractory to conventional management, including 18 who got a permanently implanted device [62]. A sub-analysis of subjects in SENZA-PPN showed 6 of 7 subjects who had PDN were responders (≥50% pain relief) and pain remitters (VAS ≤ 3.0 cm) after 12 months of 10 kHz stimulation, and 5 subjects demonstrated improvements in sensory and/or reflex testing, suggesting 10 kHz SCS could be associated with beneficial neurological effects beyond simple analgesia [63]. All study-related AEs were mild or moderate and resolved without sequalae.

More recently a prospective, multicenter RCT, SENZA-PDN, evaluated 10 kHz SCS in 216 subjects with refractive PDN who were randomized to two treatment groups, conventional medical management (CMM) and 10 kHz SCS plus CMM [88]. All subjects had pain for at least one year that was refractive to treatment with at least two pharmacologic interventions including pregabalin or gabapentin, and their pain intensity was ≥5 cm on a 10 cm VAS. Moreover, this study includes a battery of neurological assessments to monitor motor, sensory, and reflex function, and subjects were required to have stable neurological status at baseline. Neurological assessments include lower limb motor function, L1–S1 sensation to light touch, pinprick and Semmes–Weinstein 10-g monofilament sensory testing of the feet, patellar and Achilles reflexes, and Babinski response [65,88]. SENZA-PDN will also examine medication use, since previous studies have shown treating chronic pain with 10 kHz SCS is associated with decreased use of opioid analgesics [94]. This study is the largest RCT of SCS undertaken in patients with PDN, with 5 times as many patients in the treatment group as the RCT conducted with conventional SCS (Table 4) [58], and subjects will ultimately be followed for 24 months, providing important long-term data on the efficacy and safety of 10 kHz SCS [88].

The six-month results of this study were presented at the 2021 meeting of the North American Neuromodulation Society (NANS) and have been published recently [65]; the primary endpoint of this study was a composite of ≥50% pain relief and no deterioration in neurological status including motor, sensory, and reflex categories at three months [64]. The six-month results showed 86% of subjects who received 10 kHz SCS plus CMM met this endpoint, while only 5% of subjects who received CMM alone did. Overall, the mean pain relief was 76% with 10 kHz SCS and −2% in control subjects. Like the SENZA-PPN sub-analysis, the investigators reported neurologic improvements in motor, sensory, and/or reflex categories in 62% of subjects treated with 10 kHz SCS and only 3% of the control arm. Finally, 18 AEs were reported during the study in the 10 kHz SCS arm, including two explants due to infection (2.2%). The study is ongoing to determine the durability of pain relief after two years (24 months) of treatment [65].

## 6. Discussion

It is obvious from the foregoing review that there are many possible options for treating PDN, but it is also clear that the evidence supporting each of these options is not of equivalent quality. Pharmacotherapies are often the first option for treating chronic pain conditions, including PDN and other neuropathies [16], and the pivotal trials for approved drugs showed significant, but modest, effects on pain by anticonvulsants and antidepressants [28,29,30,40,41]. However, many patients do not achieve adequate pain relief with pharmacological treatments [3]. Moreover, these trials tested the efficacy or safety of these agents for 12 weeks or less, but the chronic nature of PDN requires patients to take these drugs for years. Topical capsaicin was tested longer, but had a modest benefit over placebo (7%) [50], while the clinical trials assessing tapentadol were enriched with responders before randomization, making it difficult to interpret the implications of these results for the wider population [47,48].

Among the neuromodulation treatments available, RCTs for TENS and intrathecal pain therapy are similarly limited by short follow-up intervals, small sample sizes, or both [51,52,53,54]. Conventional SCS has been tested for up to five years in PDN [58,59,60], but the number of implanted subjects was small, less than two dozen, and neither the amount of pain relief nor the proportion of responders reached 50% at these extended follow-up times. In addition, the paresthesia necessary for analgesia with conventional SCS is not tolerable for all patients, reducing the number of people who can benefit from this modality [109].

In contrast, 10 kHz SCS is paresthesia-free and is currently being tested in PDN in an ongoing, multicenter RCT much larger than the trials in conventional SCS, with over 100 patients randomized to the 10 kHz SCS arm, and these subjects will be followed for a full two years to produce high-quality data regarding the durability of effects [88]. As a chronic, incurable condition, any treatment for PDN must be durable over years. Loss of therapeutic effect due to tolerance or habituation is one of the most common reasons for failure of conventional SCS over time [85] and remains a significant challenge in its use for chronic pain [86]. Although the reasons for tolerance are poorly understood, 10 kHz SCS has been demonstrated to produce sustained back pain relief for two years [110,111,112], and SENZA-PDN is, likewise, designed to probe long-term treatment efficacy in PDN over 24 months. The results over six months have been robust with 76% mean pain relief and a responder rate of 85%, while over 60% of subjects showed neurological improvement, which has not been observed with conventional SCS or any other treatment [64,65].

Finally, complete treatment of PDN involves more than clinical outcomes such as effective, durable, and safe management of pain. A more holistic view of PDN management includes outcomes such as changes in pain medication use, health-related quality of life measures, and costs of treatment. Most of these outcomes have not been included in most clinical trials of pharmacotherapies or neuromodulation, including SCS, but SENZA-PDN addresses these areas by examining outcomes including neurological functioning, analgesic and diabetic medication use, health-related quality of life measures including sleep quality, and the cost-effectiveness of 10 kHz SCS [64,65]. However, further studies may be needed to quantitatively assess the neurological improvements following 10 kHz SCS treatment and the results also need to be reproduced in real-world setting.

It must also be noted that SCS is an invasive procedure and the adverse events associated with SCS need to be carefully weighed against the benefits before recommending the therapy. It is opined that a fraction of PDN patients may have improvement in pain symptoms within a year without any specific treatment. Therefore, it is important to carefully select the patients who are refractory to conventional medical management. In fact, the main inclusion criteria in SENZA-PDN study was pain for >1 year that is refractory to at least two pharmacologic treatments.

## 7. Conclusions

PDN is a common complication of DM that is associated with a significant decline in quality of life. This review has summarized the primary evidence for these therapies including efficacy and AEs. Although all have been shown to adequately address pain in patients with PDN, 10 kHz SCS provides evidence in a large RCT for clinically significant pain relief in addition to improvement in neurologic symptoms. SCS therapies along with pharmacological interventions provide growing armamentarium for pain management in PDN patients, in conjunction with the current state of the art in clinical management of diabetic patients.

## Figures and Tables

**Table 1 biomedicines-09-00573-t001:** Therapeutic options for painful diabetic neuropathy.

Pharmacotherapies	Neuromodulation
**Anti-convulsants** Pregabalin Gabapentin	Intrathecal pain therapy
	Transcutaneous electrostimulation *
	Tonic SCS *
Topical Capsaicin	Burst SCS *
**Opioids** Tapentadol	10 kHz SCS *
**Anti-depressants** Duloxetine Amitriptyline * Venlafaxine *	

* Not FDA approved at the time of preparation of manuscript.

**Table 2 biomedicines-09-00573-t002:** Pharmacotherapies for painful diabetic neuropathy.

Drug	Dose Range	Starting Dose	Dose Escalation	Mechanism	Side-Effects
**Pregabalin**	Up to 100 mg TID	50 mg TID	Escalate to 100 mg TID within 1 week of initiation based on tolerability	Inhibition of voltage gated calcium channels	Somnolence, blurred vision, difficulty with concentration/thinking, dry mouth, edema, weight gain Serious side-effects: Allergic reactions, suicidal thoughts, dizziness, fall and troubled breathing
**Gabapentin ***	1800 mg/day–3600 mg/day	300 mg QD	Increase to 300 mg BID and TID; then escalate dose at TID	Inhibition of voltage gated calcium channels	Dizziness, fall, somnolence, peripheral edema, and gait disturbance
**Duloxetine**	60 mg QD	≤60 mg/day	N/A	Serotonin/norepinephrine reuptake inhibitor	Nausea, somnolence, decreased appetite, constipation, fatigue, and dry mouth Serious side-effects: Suicidal thoughts, bleeding, and blurred vision
**Topical Capsaicin**	1–4 applications of 8% patch for 30 min every 3 months	1–4 applications of 8% patch for 30 min	Can be repeated not more than every 3 months	TRPV1 agonist	Application site erythema, pain, and pruritus Serious side-effects: Allergic reaction, dizziness, trouble breathing
**Tapentadol**	100 mg/day–250 mg/day (500 mg/day MRD)	50 mg BID	Individually titrated by 50 mg no more than twice daily every three days	µ-opioid receptor agonist and norepinephrine reuptake inhibitor	Nausea, constipation, dizziness, headache, and somnolence Serious side-effects: fall, seizures and difficult breathing
**Amitriptyline ^#^**	10 mg/day–150 mg/day in the night. Maximum dose 150 mg	10–25 mg/day in the night	Increase by 10–25 mg/day every 3–7 days as tolerated	Serotonin/norepinephrine reuptake inhibitor	GI issues, orthostatic hypotension, dry mouth, urinary retention, constipation and QTc prolongation Serious side-effects: Arrhythmias, suicidal thoughts and muscle cramps
**Venlafaxine ^#^**	150 mg/day–225 mg/day	75 mg/day in 2–3 divided doses	Increments of 75 mg/day every 4 days or more as tolerated	Serotonin/norepinephrine reuptake inhibitor	Nausea, somnolence, insomnia and dyspepsia Serious side-effects: dizziness, fall, hallucinations and increased heart rate

Information included in the table is summarized from FDA-approved package inserts and literature [21,22]. Side-effects profile listed in the table is not exhaustive and only includes commonly seen events. * Gabapentin approved for postherpetic neuralgia. ^#^ Not FDA approved for PDN.

**Table 3 biomedicines-09-00573-t003:** Clinical trial data for pharmacotherapies of PDN.

Drug	Study Design	N	Time to Last Follow-up	Mean Pain Relief at Last Follow-up	Responder Rate (≥50% Pain Reduction) at Last Follow-up	Secondary Outcomes
**Pregabalin**	Double-blind randomized, placebo controlled parallel-group trial [28]	75	8 weeks	38%	40%	Statistically significant improvements seen in sleep and SFMPQ scores in pregabalin treated subjects. Other outcomes included PGIC, CGIC and SF-36
**Gabapentin**	Multicenter double-blind randomized, placebo controlled trial [29]	82	8 weeks	39%	Not reported	Gabapentin treated subjects showed statistically significant improvements in SF-36 scores, sleep and in profile of mood states
**Duloxetine**	Double-blind randomized, placebo controlled parallel-group trial [30]	116	12 weeks	64% and 68% for 60 mg and 120 mg dose groups, respectively	50% and 39% respectively	Duloxetine treated subjects showed statistically significant improvements in SFMPQ scores, PGI interference and in BPI interference
**Topical Capsaicin**	Multicenter double-blind randomized, placebo controlled trial [31]	186	12 weeks	28%	19%	Topical capsaicin treated subjects mainly showed improvements in sensation
**Tapentadol**	Double-blind randomized, placebo controlled withdrawal study [32]	166	12 weeks	Subjects continuing on tapentadol had significantly less pain (26%)	40%	Tapentadol treated subjects showed statistically significant improvements in PGI, SFMPQ scores and BPI interference
**Amitriptyline**	Double-blind, randomized, cross-over active controlled trial [33]	33	6 weeks	40%	55%	No significant difference was noted between amitriptyline and duloxetine groups in secondary measures including MPQ scale and sleep improvements
**Venlafaxine**	Multicenter double-blind randomized, placebo controlled trial [34]	82	6 weeks	50%	56%	Improvements were noted on clinician rated and patient rated global improvement

**Table 4 biomedicines-09-00573-t004:** Neuromodulation therapies for painful diabetic neuropathy.

Therapy	Study Design	N at Last Follow Up	Time to Last Follow Up	Mean Reduction in Pain at Last Follow-up Compared to Baseline	Responder Rate ^e^	Quality of Life Improvements	Neurological Improvements	Adverse Events
**TENS**								
**TENS vs. high frequency EMS** [51]	Pilot RCT	12 TENS | 13 HF-EMS	3 days	NR	25% vs. 69%, respectively	NR	NR	Muscular discomfort (HF-EMS)
**PEMF** [52]	Multicenter RCT	90 in PEMF group and 104 in sham group	3 months	No significant difference between treatment and sham groups. PDN related PGIC was significantly higher (44%) in PEMF group compared to sham group (31%; *p* = 0.04)	NR	itching scores	NR	Allodynia
**FREMS** [53]	Multicenter RCT	39 in the FREMS group and 36 in the placebo group	3 cycles ^d^	NR	Day = 50% vs. 23%| Night = 54% vs. 24%, respectively	NR	Significant increase in cold sensation threshold was seen in the FREMS group	Mild, transient burning sensation at electrode site in the FREMS group
**Intrathecal pain therapy ^a^**								
**IT Ziconotide** [54]	Double-blind RCT	169 in ziconotide group and 86 in placebo group	6 days	31% in ziconotide group and 6% in placebo group	34% in ziconotide group and 13% in placebo group ^f^	Walking ability was improved	NR	Nausea; hypotension; dizziness; somnolence; urinary retention; asthenia; amblyopia; nystagmus; abnormal gait; confusion
**Conventional SCS**								
**Tesfaye et al.** [55]	Prospective, single-arm	8	14 months	Background pain: 70% Peak pain: 75%	Background pain: 86% Peak pain: 71%	Improvements in exercise threshold were noted	NR	Loss of benefit; infection; hematoma
**De Vos et al.** [56]	Prospective, open-label	9	30 months	71%	88%	N/A	NR	Lead revision; infection
**Pluijms et al.** [57]	Prospective, open-label	12	12	Day time pain, 52%; Night-time pain, 41%; Peak pain, 22%	Overall success: 67%	Improvements in SF-36 PCS; Sleep NRS scores were reported	None	Lead revision
**Slangen et al.; van Beek** [58,59]	Multicenter RCT	15 in SCS group and 14 in control group	Primary endpoint: 6 months SCS group follow-up: 24 months	SCS group: At 6 months: Day time pain, 55%; Night-time pain, 48% At 24 months: Day time pain, 45%; Night-time pain, 48%	SCS group: At 6 months: Day time pain, 53%; Night-time pain, 47% At 24 months: Day time pain, 47%; Night-time pain, 35%	Improvements in neuropathic pain scale, EQ5D, SF-36, Study sleep scale were reported	NR	Subdural hematoma causing death; infection requiring explant; lead revision; IPG replacement
**van Beek et al. 5-years follow up** [60]	Prospective multicenter long-term follow-up study	40	60	At 60 months, Day time pain, 36%; Night-time pain, 31%	At 60 months, Day time pain, 36%; Night-time pain, 32%	N/A	NR	Infection; pocket pain; uncomfortable stimulation; battery relocation; lead revision; lead replacement; battery replacement
**de Vos et al.** [61]	Open-label RCT	40 in the SCS group and 20 in the control group	6 months	55% in the SCS group vs. 0% in control group	60% in the SCS group vs. 1% in control group	Improvements in MPQ QoL; EQ5D and PGIC were reported in the SCS group	NR	Infection; pocket pain; lead migration
**High-frequency SCS**								
**SENZA-PPN** [62]	Prospective, multicenter	18	12 months	64%	69%	Improvements in PDI; SF-MPQ-2; GAF; PSQ-3 were reported	Sensory improvements were noted	Pain in extremity; implant site seroma; wound infection; implant site dehiscence
**PDN sub-analysis** [63]	Post-hoc analysis of SENZA-PPN study	7	12 months	74%	86%	Improvements in PDI; SF-MPQ-2; GAF; PSQ-3 were reported	Sensory and reflex improvements were noted	Pain in extremity; implant site seroma ^g^
**SENZA-PDN ^b^** [64,65]	Multicenter RCT	87	6 months	76% in 10 kHz SCS group vs. 1% in control group	85% in 10 kHz SCS group vs. 5% in control group	Improvements in PSQ-3, EQ5D, GAF and SF-12 were reported in the 10 kHz SCS group	Sensory, motor, reflex improvements were noted	Explant due to infection (#2)

a—IT pain therapy tested in subjects with non-malignant pain, including neuropathic pain; b—Ongoing study; d—Results after 3 cycles of treatment; e—Calculated using received treatment (RT) population; f—Response defined as pain relief ≥ 30%; g—Subset of AEs reported in SENZA-PPN. Definitions: TENS = Transcutaneous electrical nerve stimulation; EMS = External muscle stimulation; PEMF = Pulsed electromagnetic fields; FREMS = Frequency-modulated electromagnetic neural stimulation; IT = Intrathecal; SCS = Spinal cord stimulation; NR = Not reported; NS = Not significant; SF-36 PCS = Short Form-36 Physical Component Score; SPI-9 = Sleep Problems Summary 9; MPQ QoL = McGill Pain Questionnaire Quality of Life Score; EQ5D = EuroQoL 5D; PGIC = Patient Global Impression of Change; PDI = Pain Disability Index; SF-MPQ-2 = Short-Form McGill Pain Questionnaire; SF-MPQ-2 = Short-Form McGill Pain Questionnaire; GAF = Global Assessment of Functioning; PSQ-3 = Pain and Sleep Questionnaire.

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
