# Peer review of "Treatment of Painful Diabetic Neuropathy—A Narrative Review of Pharmacological and Interventional Approaches"

_biomedicines, 2021, doi:10.3390/biomedicines9050573_

Round 1
Reviewer 1 Report
Topic is important and contents are new and interesting to read. The information is overall updated and worthy of publication. There are several concerns that should be addressed for the improvement of the content.
- Table 1, list of options needs to be classified not by the name of compounds but the category of drugs to follow the statement in the text. i.e. anti-convulsants (pregabalin, gabapentin, etc), anti-depressant (duloxetine, venlafaxine, amitriptyline etc), neuromodulation (~~~~~), decompression surgery etc.
- English text needs to be carefully edited in some places (see Page 2 L1-3 from the bottom), and more.
- Table 2; Clinical symptoms of painful diabetic neuropathy. This table does not make sense. Are they symptoms which need to be treated or uncomfortable for patients?
- Table 3 Pharmacotherapies for PDN. Please include “fall” in the column of side effects.
- As described, opioids are not recommended to use for PDN. This should be more emphasized, because it is hard to get rid of this addiction which could have been avoided.
- Non-pharmacological treatments: neuromodulation. This part of statements should be carefully and described in a well-balanced manner. For the decision of treatment, benefits should overcome the demerit. It seems that adverse effects are not trivial. Table 5 should present more clearly the results of the trial. The values in Mean Pain Relief at Last Follow up are confusing.
Author Response
- Table 1, list of options needs to be classified not by the name of compounds but the category of drugs to follow the statement in the text. i.e. anti-convulsants (pregabalin, gabapentin, etc), anti-depressant (duloxetine, venlafaxine, amitriptyline etc), neuromodulation (~~~~~), decompression surgery etc.
Authors’ response: We have revised Table 1 as suggested.
- English text needs to be carefully edited in some places (see Page 2 L1-3 from the bottom), and more.
Authors’ response: We thank the reviewer for feedback. We have carefully edited the text for language and grammatical errors.
- Table 2; Clinical symptoms of painful diabetic neuropathy. This table does not make sense. Are they symptoms which need to be treated or uncomfortable for patients?
Authors’ response: We have removed Table 2 in response to reviewer’s comment.
- Table 3 Pharmacotherapies for PDN. Please include “fall” in the column of side effects.
Authors’ response: We have added serious side effects and ‘fall’ in the column of side effects as suggested.
- As described, opioids are not recommended to use for PDN. This should be more emphasized, because it is hard to get rid of this addiction which could have been avoided.
Authors’ response: We agree with the reviewer that it is hard to get rid of opioid addiction and could be avoided. In response to reviewer’s suggestion, we have revised our opioids section and added an advisory sentence on avoiding opioids. Please refer to page 9, paragraph 1 for details.
- Non-pharmacological treatments: neuromodulation. This part of statements should be carefully and described in a well-balanced manner. For the decision of treatment, benefits should overcome the demerit. It seems that adverse effects are not trivial. Table 5 should present more clearly the results of the trial. The values in Mean Pain Relief at Last Follow up are confusing.
Authors’ response: We thank the reviewer for feedback. We have revised our discussion section and added a note on weighing the risks and benefits of SCS before recommending the therapy. Please refer page 18, paragraph 4 for details.
We have also updated Table 5 (now Table 4) and clarified the information from the trial. Please refer to revised Table 4 for details.
Reviewer 2 Report
My review comments are summarized below, in order of their appearance, not necessarily importance.
I am not familiar with the term “narrative review.” This paper includes no review methodology, and I assume the term narrative simply implies a broad perspective review based mainly on personal experience. To me, this paper reads more like a book chapter than a journal article.
Accompanying this manuscript, I was provided four accepted articles from the current issue, including one on the “Efficacy and safety of 10kHz spinal cord stimulation for the treatment of chronic pain: a systematic review and narrative synthesis of real-world retrospective studies.” Unlike the present manuscript, that review included a methods section describing the literature search, study eligibility, reviewers, and data extraction methodology, a reproducible review that reduces bias.
Aside from my methodological concerns, I have no issues with the background, clinical descriptions, or general management discussions. The descriptions are succinct and appropriately referenced.
The pharmacotherapy section is easy to read, presented in a manner consistent with a broad overview. The information included some useful consensus recommendations from organizations like the AAN and ADA. The summary tables were very helpful. This applies to the sections describing the use of anticonvulsant medications and, to a lesser extent, antidepressants and capsaicin.
My experience with the use of opioid agonist may differ somewhat from the authors and is more consistent with the recommendations of the ADA not to recommend this medication as a first- or second-line treatment. Despite noting these concerns in the last paragraph of the opioid discussion, the authors provide dosing information for tapentadol, with instructions to titrate to a dose providing adequate analgesia while avoiding AEs, providing a maximum daily dose. This juxtaposition left me unclear what their recommendation actually is regarding the use of opioid agonists.
I’m not familiar with the authors or their work. However, in my initial read through the manuscript, I sensed a commercial bias toward the non-pharmacological treatments. The material included optimistic predictions about the future application of neuromodulation, a different tone than in the preceding material, resembling a grant proposal or an infomercial for 10 kHz neuromodulation.
That aside, the sections involving TENS and intrathecal pain therapy were consistent in tone with the preceding sections. Because this is a narrative review, I cannot evaluate how the studies were selected.
The sections on conventional and burst spinal cord stimulation are well-written, but the level of detail exceeds that of earlier sections. I am unable to judge the quality of the citation selections.
In my view, the section on “high-frequency SCS” stands out from the other sections, in terms of its length, degree of detail, the inclusion of animal studies, and subjective comments (such as its ability to produce “deep and durable paresthesia-free pain relief for chronic neuropathic pain”). I don’t challenge the conclusions, which are based on material I cannot review, only the methodology used to support the conclusions.
In this context, the greatest attention was given to the SCNZA-PPN studies. The pilot study involving 26 subjects with peripheral polyneuropathy “refractory to conventional management,” and the more recent prospective study involving 226 participants with painful diabetic neuropathy. The favorable results presented at a recent meeting included ≥ 50% pain relief with no adverse neurologic signs among 86% of subjects in the 10-kHz SCS group versus 5% of those receiving CCM alone, after six months of follow-up. I admit surprise that only 5% of those receiving CCM alone exceeded the 50% criteria of pain relief. Again, this is material I am unable to critique, as it has not been published as of my review.
The first reference above describes the design of the trial as a comparison of subjects receiving conventional medical management, with or without 10-Hz SCS (1:1 assignment). The results of that trial appear in the proceedings of a 2021 virtual meeting of the North American Neuromodulation Society, something I cannot access. When I Googled the title, however, the first item was a description of the results provided by Nevro Corp., the sponsor of the device, under Investor News Details.
The SCNZA-PPN study raises questions that I cannot address. Most physicians involved in pain management are familiar with the substantial placebo effect associated with any surgical procedure. There may be RCTs comparing the use of different stimulation frequencies that could address those issues. Unfortunately, I am not familiar with such studies.
The discussion section is well-written, but the feeling of subtle commercial bias is difficult to overlook. Namely, all of the treatments available are critically critiqued, identifying the shortcomings of each, the exception being the 10 kHz SCS, a treatment described as paresthesia-free and being tested in diabetic peripheral neuropathy in large RCTs.
This enthusiastic support of a methodology currently being studied seems out of place in a critical review article. By the third sentence of the conclusion section, only 10 kHz SCS is said to provide evidence in a large RCT for clinically significant pain relief in addition to improvement in neurologic symptoms (albeit not yet available for review). These two sections read like a promotional release for 10 kHz SCS, not a review article.
Missing in the discussion is clarification on the prevalence of painful diabetic neuropathy refractory to conventional treatment. The epidemiology of painful diabetic neuropathy is complicated. I agree that about 20% of patients with diabetic neuropathy fulfill the criteria for a painful DN, but would add that many of these patients describe symptomatic improvement within a year or so, some without any specific treatment. This reduction in pain often occurs in the setting of progressive diabetic neuropathy, making detailed clinical descriptions important, something that cannot be derived from an abstract summary.
Author Response
My review comments are summarized below, in order of their appearance, not necessarily importance.
I am not familiar with the term “narrative review.” This paper includes no review methodology, and I assume the term narrative simply implies a broad perspective review based mainly on personal experience. To me, this paper reads more like a book chapter than a journal article.
Authors’ response: As reviewer rightly pointed out, the term narrative simply implies a broad perspective review based mainly on personal experience. However, we also had a broad methodology for selecting the articles. We mainly selected the pivotal RCTs that either helped the treatment get FDA approval or established the efficacy of the treatment. In response to reviewer’s suggestion, we have included methodology for selection of articles. Please refer to page 4, paragraph 3 for details.
Accompanying this manuscript, I was provided four accepted articles from the current issue, including one on the “Efficacy and safety of 10kHz spinal cord stimulation for the treatment of chronic pain: a systematic review and narrative synthesis of real-world retrospective studies.” Unlike the present manuscript, that review included a methods section describing the literature search, study eligibility, reviewers, and data extraction methodology, a reproducible review that reduces bias.
Authors’ response: As described above, we have included the methodology for selection of articles.
Aside from my methodological concerns, I have no issues with the background, clinical descriptions, or general management discussions. The descriptions are succinct and appropriately referenced.
Authors’ response: We thank the reviewer for the feedback.
The pharmacotherapy section is easy to read, presented in a manner consistent with a broad overview. The information included some useful consensus recommendations from organizations like the AAN and ADA. The summary tables were very helpful. This applies to the sections describing the use of anticonvulsant medications and, to a lesser extent, antidepressants and capsaicin.
Authors’ response: We thank the reviewer for the feedback.
My experience with the use of opioid agonist may differ somewhat from the authors and is more consistent with the recommendations of the ADA not to recommend this medication as a first- or second-line treatment. Despite noting these concerns in the last paragraph of the opioid discussion, the authors provide dosing information for tapentadol, with instructions to titrate to a dose providing adequate analgesia while avoiding AEs, providing a maximum daily dose. This juxtaposition left me unclear what their recommendation actually is regarding the use of opioid agonists.
Authors’ response: We agree with the reviewer that opioids should not be recommended as a first or second-line treatment. We have revised the section on opioids and emphasized the need to avoid opioids for the management of pain. Please refer to page 9, paragraph 1 for details.
I’m not familiar with the authors or their work. However, in my initial read through the manuscript, I sensed a commercial bias toward the non-pharmacological treatments. The material included optimistic predictions about the future application of neuromodulation, a different tone than in the preceding material, resembling a grant proposal or an infomercial for 10 kHz neuromodulation.
Authors’ response: We would like to emphasize that the goal of this review was to present results from pivotal RCTs and discuss efficacy along with the observed adverse events. There was no intention of bias toward non-pharmacological treatments or towards 10 kHz neuromodulation. We believe that the SENZA-PDN study was well designed and data from study presented at NANS2021 and now published in JAMA Neurology (reference #106) needed special discussion to facilitate future research.
That aside, the sections involving TENS and intrathecal pain therapy were consistent in tone with the preceding sections. Because this is a narrative review, I cannot evaluate how the studies were selected.
Authors’ response: We thank the reviewer for the feedback. We have now provided the rationale for selection of articles.
The sections on conventional and burst spinal cord stimulation are well-written, but the level of detail exceeds that of earlier sections. I am unable to judge the quality of the citation selections.
Authors’ response: We thank the reviewer for feedback. As detailed in our brief section on methodology, we selected the pivotal RCTs and discussed their findings. Level of detail was mainly based on the studies and key takeaways.
In my view, the section on “high-frequency SCS” stands out from the other sections, in terms of its length, degree of detail, the inclusion of animal studies, and subjective comments (such as its ability to produce “deep and durable paresthesia-free pain relief for chronic neuropathic pain”). I don’t challenge the conclusions, which are based on material I cannot review, only the methodology used to support the conclusions.
Authors’ response: At the time of submission of the manuscript, the findings discussed from pivotal study on high-frequency SCS were from a conference presentation. However, the study is now published in JAMA Neurology (reference #106). We believe it as a well-designed study, with decent sample size (N=216) and is expected to follow-up the patients over longer term (24 months), which is mainly the reason for the length and degree of details in the discussion. We respectfully disagree that the statement on paresthesia-free pain relief is a subjective comment. We would like to point out that paresthesia-free pain relief is a characteristic feature of high-frequency SCS and has been documented in several studies.
In this context, the greatest attention was given to the SCNZA-PPN studies. The pilot study involving 26 subjects with peripheral polyneuropathy “refractory to conventional management,” and the more recent prospective study involving 226 participants with painful diabetic neuropathy. The favorable results presented at a recent meeting included ≥ 50% pain relief with no adverse neurologic signs among 86% of subjects in the 10-kHz SCS group versus 5% of those receiving CCM alone, after six months of follow-up. I admit surprise that only 5% of those receiving CCM alone exceeded the 50% criteria of pain relief. Again, this is material I am unable to critique, as it has not been published as of my review.
Authors’ response: As noted above, the SENZA-PDN study has been published in JAMA Neurology. We have included the reference (#106) in our revised manuscript.
The first reference above describes the design of the trial as a comparison of subjects receiving conventional medical management, with or without 10-Hz SCS (1:1 assignment). The results of that trial appear in the proceedings of a 2021 virtual meeting of the North American Neuromodulation Society, something I cannot access. When I Googled the title, however, the first item was a description of the results provided by Nevro Corp., the sponsor of the device, under Investor News Details.
Authors’ response: Please see above.
The SCNZA-PPN study raises questions that I cannot address. Most physicians involved in pain management are familiar with the substantial placebo effect associated with any surgical procedure. There may be RCTs comparing the use of different stimulation frequencies that could address those issues. Unfortunately, I am not familiar with such studies.
Authors’ response: We agree that SENZA-PDN study was not designed to evaluate the placebo effect associated with the surgical procedure. However, we would like to point out that placebo effects can commonly last for shorter follow-up period and less likely to last over 12 months, which is the follow-up period for SENZA-PDN study. Moreover, prospective studies in the field of SCS are commonly intended to have follow-up of 12 to 24 months and there can be serious ethical and practical concerns in designing a sham control study in patients. Concerns on placebo effects can be at least partly addressed by studies in animal models and we have cited the sham-controlled studies (reference #96 and #105) in rats that documented objective responses following 10 kHz SCS treatment.
The discussion section is well-written, but the feeling of subtle commercial bias is difficult to overlook. Namely, all of the treatments available are critically critiqued, identifying the shortcomings of each, the exception being the 10 kHz SCS, a treatment described as paresthesia-free and being tested in diabetic peripheral neuropathy in large RCTs.
Authors’ response: We respectfully disagree with the reviewer that we had any bias towards 10 kHz SCS treatment. In our opinion, SENZA-PDN study was well designed and deserved the emphasis. In response to reviewer’s comment, we have revised our manuscript, provided rationale for articles selected for discussion and added the short comings of SENZA-PDN. Please refer to Page 18, paragraph 3 for details.
This enthusiastic support of a methodology currently being studied seems out of place in a critical review article. By the third sentence of the conclusion section, only 10 kHz SCS is said to provide evidence in a large RCT for clinically significant pain relief in addition to improvement in neurologic symptoms (albeit not yet available for review). These two sections read like a promotional release for 10 kHz SCS, not a review article.
Authors’ response: As noted above, we have revised the manuscript and provided critical review of SENZA-PDN (page 18, paragraph 3). However, we stand by our conclusion that only SENZA-PDN is the large RCT documenting significant improvement in neurologic symptoms and expected to have long-term follow-up (24 months). Other pivotal studies either had a smaller sample size, did not document changes in neurologic symptoms or had a short follow-up (<3 months), compared to SENZA-PDN (reference #106).
Missing in the discussion is clarification on the prevalence of painful diabetic neuropathy refractory to conventional treatment. The epidemiology of painful diabetic neuropathy is complicated. I agree that about 20% of patients with diabetic neuropathy fulfill the criteria for a painful DN, but would add that many of these patients describe symptomatic improvement within a year or so, some without any specific treatment. This reduction in pain often occurs in the setting of progressive diabetic neuropathy, making detailed clinical descriptions important, something that cannot be derived from an abstract summary.
Authors’ response: We have added a clarification that some patients describe symptomatic improvement within a year and that patients need to be carefully selected given the invasiveness of the procedure. Please refer to page 18, paragraph 4 for details.
Reviewer 3 Report
The authors review available pharmacological and non-pharmacological options for treatment of painful diabetic neuropathy (PDN). The review is succinct and provides comprehensive summary of the topic as well as the summary of the primary evidence for the available therapies for PDN.
Author Response
The authors review available pharmacological and non-pharmacological options for treatment of painful diabetic neuropathy (PDN). The review is succinct and provides comprehensive summary of the topic as well as the summary of the primary evidence for the available therapies for PDN.
Authors’ response: We thank the reviewer for the feedback.